# A multimodal dataset for precision oncology in head and neck cancer

Marion Dörrich[1], Matthias Balk[2,3], Tatjana Heusinger[2,4], Sandra Beyer[2,5], Hamed Mirbagheri[1], David J. Fischer[1], Hassan Kanso[2,3], Christian Matek [3,6], Arndt Hartmann[3,6,7], Heinrich Iro[2,3], Markus Eckstein [3,6,7,8], Antoniu-Oreste Gostian[2,3,4,7,8] & Andreas M. Kist [1,8] ✉

Head and neck cancer is a common disease and is associated with a poor prognosis. A promising approach to improving patient outcomes is personalized treatment, which uses information from a variety of modalities. However, only little progress has been made due to the lack of large public datasets. We present a multimodal dataset, HANCOCK, that comprises monocentric, real-world data of 763 head and neck cancer patients. Our dataset contains demographical, pathological, and blood data as well as surgery reports and histologic images, that can be explored in a low-dimensional representation. We can show that combining these modalities using machine learning is superior to a single modality and the integration of imaging data using foundation models helps in endpoint prediction. We believe that HANCOCK will not only open new insights into head and neck cancer pathology but also serve as a major source for researching multimodal machine-learning methodologies in precision oncology.

Head and neck cancer is the seventh most common malignancy worldwide[1]. Patients diagnosed with head and neck cancer have a poor prognosis[2]. Despite recent advances in diagnostics and treatments, such as immunotherapy, the 5-year survival ranges only between 25% and 60%[3]. The most common head and neck cancer develops in several locations, e.g., the oral cavity, pharynx, or larynx, and is derived from squamous cells, i.e., originates from the mucosal epithelium lining the inner areas of these sites. The cancer often spreads to regional lymph nodes, which further worsens the prognosis of affected patients[4].

After assessing the medical history and physical examination, a panendoscopy with biopsy is usually performed to confirm the diagnosis. The pathological analysis of tissue samples is crucial for determining the histological entity. In addition, lymph nodes are examined for possible metastases. Surgery is one of the most important pillars of treatment for head and neck cancer. Local surgery is often sufficient for lower-stage cancer, while adjuvant treatment such as radiotherapy or radiochemotherapy is required for higher stages[5]. Despite many advances in diagnostics, the treatment choice still depends mainly on the stage of the disease, that is mainly determined by the size of the tumor[5,6]. However, research showed that cancer is highly diverse among patients[7] and therefore requires precision oncology. The key to this personalized treatment is the establishment of reliable and predictive biomarkers. Initiatives such as The Cancer Genome Atlas (TCGA) have already achieved a better understanding of the genetic and molecular characteristics of many types of cancer[8].

[1]Department Artificial Intelligence in Biomedical Engineering, Friedrich-Alexander-Universität Erlangen-Nürnberg, Erlangen, Germany. [2]Department of Otolaryngology - Head and Neck Surgery, University Hospital Erlangen, Friedrich-Alexander-Universität Erlangen-Nürnberg, Erlangen, Germany. [3]Bavarian Cancer Research Center (BZKF), Erlangen, Germany. [4]Department of Otorhinolaryngology, Head and Neck Surgery, Merciful Brothers Hospital St. Elisabeth, Straubing, Germany. [5]Department of Oral and Maxillofacial Surgery, University Hospital Erlangen, Friedrich-Alexander-Universität Erlangen-Nürnberg, Erlangen, Germany. [6]Institute of Pathology, University Hospital Erlangen, Friedrich-Alexander-Universität Erlangen-Nürnberg, Erlangen, Germany. [7]Comprehensive Cancer Center Erlangen-EMN (CCC ER-EMN) and Comprehensive Cancer Center Alliance WERA (CCC WERA), Erlangen, Germany. [8]These authors contributed equally: Markus Eckstein, Antoniu-Oreste Gostian, Andreas M. Kist. ✉e-mail: andreas.kist@fau.de

However, very few biomarkers are currently used in routine head and neck cancer treatment. A positive prognostic biomarker is the association with human papillomavirus (HPV) in oropharyngeal carcinomas[9]. Ongoing research aims to explore if their treatment can be de-escalated to reduce toxicity[10]. Furthermore, the expression of programmed death ligand 1 (PD-L1) can be assessed to identify patients who may benefit from immune checkpoint inhibitors such as pembrolizumab, and remains the only applied predictive biomarker for now[11]. However, more reliable biomarkers need to be established to enable a truly personalized treatment. Although information from a large variety of sources is routinely acquired, its full potential cannot be realized for data-driven exploration yet. Careful data curation and multimodal integration are required to unravel complex data dependencies. We hypothesize that a lack of such large, multimodal, publicly available datasets hinders the research of predictive biomarkers for head and neck oncology.

To our knowledge, existing head and neck cancer datasets only have a limited number of cases or have inconsistent metadata[12–15]. For example, a study focusing on radiomics included data from 288 cases while only selecting oropharyngeal carcinomas[15]. Another dataset focusing on proteomics includes radiology and histopathology data, but is limited to 122 cases[13]. A first approach to provide multiple modalities for a larger cohort including clinical, genomic, and histopathologic data has been collected on TCGA from more than 500 cases to date, however, lacking further information such as blood samples or surgery reports, as well as specific imaging data for molecular subtypes, as well as the lack of availability for most presented modalities for each individual in the cohort[12,16]. These drawbacks lead to only limited usability of these data in a multimodal data integration approach.

To address these issues, we collected monocentric, retrospective data from more than 700 head and neck cancer patients. We built a comprehensive dataset from multimodal data, including demographics, blood data, surgery reports, pathologic data, and histologic images. These include Whole Slide Images (WSIs) with routine hematoxylin and eosin (HE) staining and Tissue Microarrays (TMAs) with staining for several immune cell populations. In this work, we aim to explore and provide reproducible strategies for multimodal integration and analysis. We further aim to predict patient outcomes using multimodal Machine Learning (ML) strategies to show the impact of multimodal data integration for head and neck oncology.

## Results

### Compilation of a multimodal dataset from a head and neck cancer cohort

Patient diagnoses and treatment decisions are rarely based on a single modality; hence, artificial intelligence (AI) models intended to assist clinicians should adopt a holistic approach, incorporating multiple data sources. Training such models requires extensive and diverse patient data, which is often scarce. To address this, we have aggregated a comprehensive dataset, HANCOCK (Head And Neck Cancer dataset), which consists of real-world data from 763 patients. In detail, we collected, cleaned, and harmonized routinely acquired monocentric data from patients diagnosed with oral cavity, oropharyngeal, hypopharyngeal, and laryngeal cancer. We integrated different modalities, including demographics, blood data, pathology reports, surgery reports, and histologic images, as shown in Fig. 1A. We provide an overview and easy, public access to the individual patient data for convenient manual exploring at www.hancock.research.fau.eu with support of the FAUDataCloud.

A core strength of HANCOCK is its rich base of imaging data: HE-stained WSIs of the primary tumor are available for 701 out of 763 patients. We also provide manual annotations of tumor regions in these WSIs, as shown in Supplementary Fig. S1. In addition, 396 HE-stained slides of adjacent lymph nodes were included. Each patient

contains at most 32 TMAs, which reflect two cores, eight stains, and two locations. Each core is stained with either HE or immunohistochemistry (IHC) markers, such as CD3 and PD-L1. Figure 1B shows exemplarily the available imaging data for a single patient. For each patient, the pathology report was included in a structured format. These cover tumor characteristics such as the primary site or grading (see Fig. 1E), crucial for selecting a suitable treatment. Additional characteristics such as tumor staging, resection margin, and infiltration depth are summarized in Supplementary Fig. S2.

As shown in Fig. 1C, 80% of the patients in the dataset are males, and 72% are former or current smokers. The median age is 61 years. Thus, our patient cohort reflects the current demographics of head and neck cancer[1], which is beneficial for generalizing our findings to a broader population. The laboratory data includes the complete blood count as well as coagulation parameters, electrolytes, renal function parameters, and C-reactive protein. Figure 1D shows for how many patients the individual parameters are available and how many of the measured blood parameters are in the normal or abnormal range.

The incorporation of treatment information and temporal event data allows an in-depth analysis of the underlying relationships. To this end, we extracted and de-identified plain text descriptions of the surgery and medical history from text documents. Figure 1F illustrates the length of surgery reports, which increases with the pathological T stage ($r = 0.404$). All German text files were translated into English to improve their accessibility (see "Methods"). OPS codes (German procedure classification) define the medical procedures applied. We also extracted ICD codes (International Statistical Classification of Diseases and Related Health Problems) of the German version ICD-10-GM from the text documents. The ICD codes allow a detailed classification of malignancies and their sites. The most frequent ICD codes were C10.8 and C32.0, as shown in Supplementary Fig. S3D. C10.8 corresponds to a malignant neoplasm in overlapping regions of the oropharynx, and C32.0 corresponds to a malignant neoplasm of the glottis[17]. We believe that ICD coded will allow easy subsampling of the full dataset, for example, only looking at subcollectives relevant to a given scientific question, such as localization and comorbidities.

In HANCOCK, each patient is tracked from the time of initial diagnosis to either the end of follow-up or death, with follow-up periods lasting as long as 14 years (see Supplementary Fig. S4). This enables the examination of temporal information, for example, in the form of treatment timelines (see Supplementary Fig. S5) and survival analyses. Figure 1G shows the overall survival of all patients in the HANCOCK dataset. Survival curves with additional information, such as the number of censored patients, can be found in Supplementary Fig. S6D, and survival curves grouped by primary site, stage, and grading are shown in Supplementary Fig. S6A–C. The 5-year survival rate in our cohort is 77.3%.

Overall, the dataset features a great variety of modalities for a large patient cohort (763 cases), which resembles the global demographics of head and neck cancer.

### Multimodal data integration allows prediction of clinical outcomes

After carefully aggregating the patient data, we were next interested in investigating the overall patient collective. To better understand the complex patient data, we encoded information from each modality individually and concatenated these embeddings into vectors, termed multimodal patient vectors, as shown in Fig. 2A. This corresponds to an early fusion approach since the modality vectors are first concatenated and then used to train a single model[18]. This aligns with our goal of creating a holistic representation of each patient (the aforementioned multimodal patient vector) that captures the interdependencies between individual modalities. In contrast, late fusion approaches are suboptimal in modeling these interconnections. For this reason, we chose to adopt the early fusion approach.

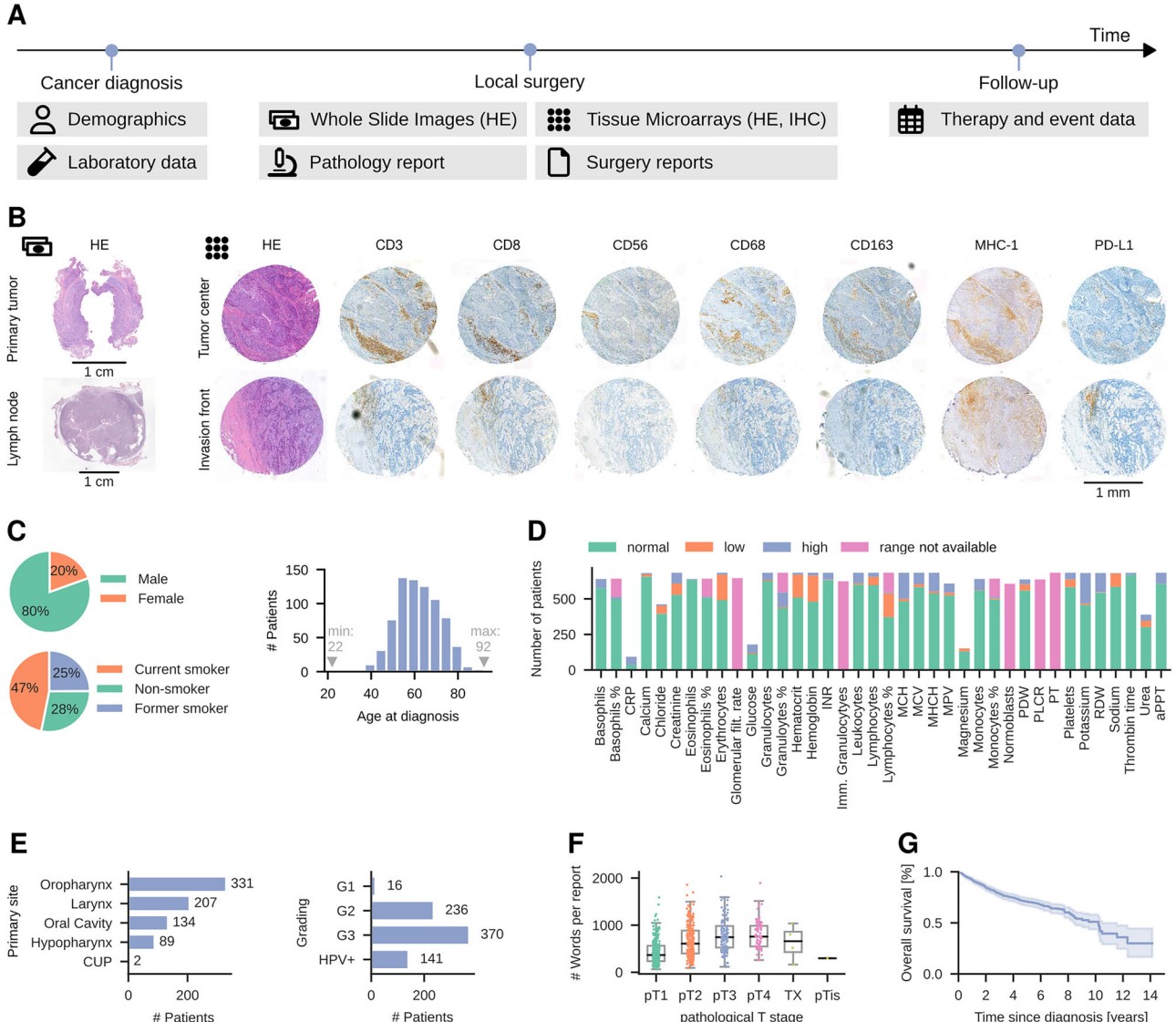

**Fig. 1 | Overview of the multimodal head and neck cancer dataset. A** Data sources. For cancer diagnosis, demographics were assessed, and blood tests were performed. In the ablative surgery, tissue samples were obtained, and the pathological report was written. The dataset also features information about the treatment choice, events, and survival. **B** Image data of a patient. Shown are Whole Slide Images of the primary tumor and lymph node with hematoxylin and eosin (HE) staining and Tissue Microarray cores from the tumor center and invasion front with HE and immunohistochemistry (IHC) staining. Scale bar as indicated (1 cm for WSI and 1 mm for TMAs). **C** Demographical data, shown as the number of patients per sex, smoking status, and age at initial diagnosis. **D** Laboratory data. Shown is the number of patients for which each parameter is available. The colors indicate values inside or outside of the normal range. **E** Primary tumor site or CUP (cancer of unknown primary) and grading from the pathology report. HPV-associated carcinoma was not graded. **F** Number of words in each German surgery report grouped by pathological T stage ($N = 742$ in total). Boxplots show Q1–Q3 interval with median, whiskers are 1.5 × the inter-quartile (Q1–Q3) range. **G** Kaplan-Meier plot of overall survival with 95% confidence interval shown as shaded error. The icons for demographics, surgery reports, therapy, and event data are CC BY licensed from Font Awesome. Source data are provided as a Source Data file.

Given the high-dimensional nature of these vectors (the multimodal patient vectors contain 104 dimensions each, see Methods and Supplementary Table S5 for an overview), these patient-centered features can hardly be examined or interpreted by humans. Therefore, we applied Uniform Manifold Approximation and Projection (UMAP) to these vectors to project them into a lower, two-dimensional space, as shown in Fig. 2B, C. In Supplementary Fig. 7, we provide a comprehensive overview of incorporated features and their distribution in the UMAP projection.

Subsequently, we sought to identify distinct patient clusters using these multimodal patient vectors. We hypothesized that similar patient groups would converge within specific areas of the two-dimensional UMAP projection. Our findings confirm this hypothesis, as we observed that patients sharing particular characteristics tended to form distinct clusters. For instance, patients diagnosed with HPV-positive oropharyngeal carcinoma often exhibited a high density of CD8 + cells, as illustrated in Fig. 2C. In addition, our analysis revealed that both CD3 + and CD8 + cell densities at the tumor center and the invasion front were notably higher in patients who did not experience recurrence compared to those who did (Supplementary Fig. S8). These observations are consistent with prior studies in head and neck oncology[9], underscoring the relevance and accuracy of the HANCOCK dataset.

We aimed to investigate whether ML models could predict clinical outcomes, i.e., recurrence and survival status, using the encoded multimodal data. We were also interested in defining different hold-out test datasets that would allow a robust estimation of a model's performance. To this end, we defined three data splits that divide the cases into one training and one test set. We hypothesize that the performance of

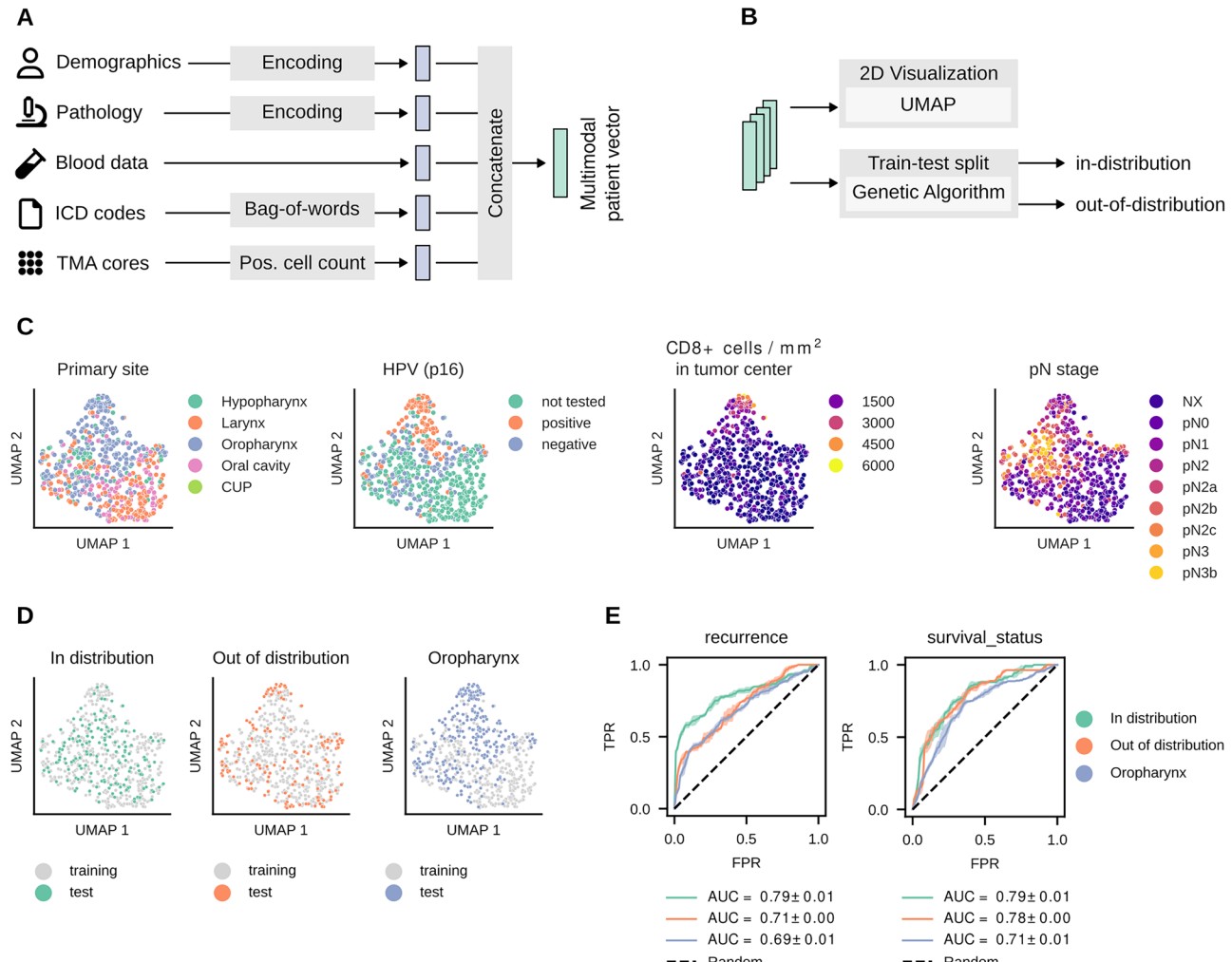

**Fig. 2 | Multimodal embeddings. A** For each patient, information from distinct modalities were encoded and concatenated to multimodal patient vectors. **B** We applied Uniform Manifold Approximation and Projection (UMAP) to visualize the vectors in 2D, and we implemented a genetic algorithm to create two test datasets, one in the distribution of the training data and one out of the distribution. **C** Visualization of two-dimensional embeddings, colored by features of the encoded data. **D** UMAP plots of three different train-test splits (**E**) Receiver-operating characteristics (ROC) curves of a Random Forest classifier for the three splits and two prediction tasks. The mean values and standard deviations of the ROC curves and Area Under the Curve (AUC) scores are shown. The colors correspond to the different splits in (**D**). The icons for demographics and ICD codes are CC BY licensed from Font Awesome. Source data are provided as a Source Data file.

models can be over- or under-estimated depending on how similar the test data is to the training data, especially in a complex, high-dimensional, and multimodal setting, as in our case. To address and investigate this issue, we implemented a genetic algorithm to automatically define two dataset splits based on multidimensional features. The algorithm uses evolutionary optimization to find (i) cases that follow the overall distribution ("in distribution") or (ii) cases that lie outside the distribution and are maximally dissimilar to each other ("out of distribution"). In both settings, the genetic algorithm preserves the distribution of target classes (recurrence and survival status) in the resulting training and test sets, which is important for model evaluation[19]. The respective class distributions are shown in Supplementary Fig. S9C, D. In addition, we defined a third split where all patients with a carcinoma located in the oropharynx were assigned to the test dataset, rendering it very dissimilar and biased to the training data. These three training/test data splits are highlighted in the UMAP representation in Fig. 2D.

Next, we trained an ML model, namely a Random Forest classifier, to predict the recurrence and survival status of each patient by using the multimodal patient vectors as inputs. We opted for the Random Forest classifier, as our results based on systematic testing together with random hyperparameter search indicated that other ML models, such as Adaboost and Support Vector classifiers, either had only similar or worse performance, respectively (Supplementary Tables S6 and S7). Respective hyperparameters are shown in the provided code (see "Methods"). Figure 2E shows the performance of the classifiers for the previously mentioned train-test splits (see Fig. 2D for reference). As expected, the model had difficulty predicting patient outcomes for the test dataset consisting of cases with oropharyngeal carcinoma, a primary site that the model has not seen during training. This is highlighted by the lowest Area Under the Curve (AUC) score as shown in Fig. 2E compared to the other test sets. In accordance with our hypothesis, the classification performance was higher for the "in distribution" than the "out of distribution' test dataset as shown in Fig. 2E. Overall, we can provide evidence that multimodal ML models follow expected ML behavior and were able to successfully estimate the prognosis of patients, achieving a maximum average AUC score of 0.79 for both recurrence and survival prediction.

## Multimodal multiple instance learning allows the integration of imaging data

In the aforementioned patient vectors, we focused on histopathology and TMA-derived information. However, the direct use of the

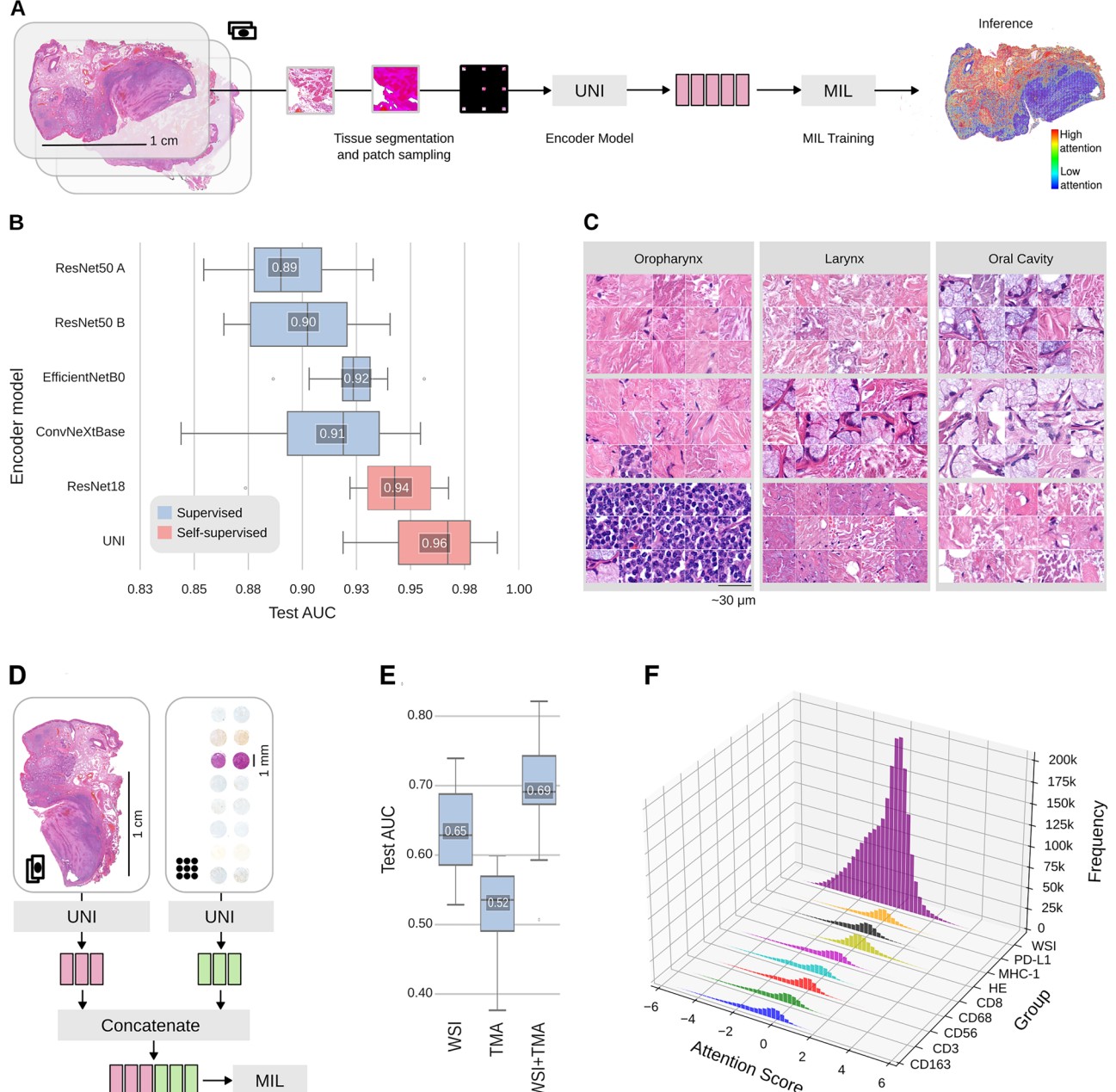

**Fig. 3 | Multimodal multiple instance learning allows the prediction of targets using imaging data. A** Multiple instance learning (MIL) pipeline. Tissue in WSIs is segmented and subsequently sampled in patches. These patches are encoded, for example, using the UNI architecture[22], and used as input for MIL together with a specified target. Using the CLAM framework[20], we can retrieve attention scores (blue: low attention, red: high attention). Scale bar is 1 cm for the WSI and its attention-labeled counterpart. **B** Slide-level AUC values for localization prediction on the test dataset ($N = 10$ each). Color-coded for supervised (blue) and self-supervised (red) encoding backbones. All backbones are based on convolutional neural networks, whereas UNI is based on vision transformers. Boxplots show

Q1–Q3 interval with median, whiskers are 1.5 × the inter-quartile (Q1–Q3) range. **C** Most attended patches for three test WSIs for all localizations tested (oropharynx, larynx and oral cavity). Note the presence of gland tissue in oral cavity-derived samples. Scale bar indicates 30 μm. **D** Multimodal integration of different imaging data sources. We use separate encodings for WSIs (pink) and TMAs (green) using the UNI encoder for MIL. **E** Slide-level AUC values for survival prediction on the test dataset ($N = 10$ each). Boxplots show Q1–Q3 interval with median, whiskers are 1.5 × the inter-quartile (Q1–Q3) range. Scale bar for WSI indicates 1 cm, for TMAs 1 mm. **F** Attention scores and their frequency across information-containing groups and modalities for the test dataset. Source data are provided as a Source Data file.

underlying imaging data has not been explored. Two main questions arose: (i) how can we efficiently encode the information of large image datasets and determine relevant content important for decision-making and (ii) integrate multiple imaging sources for endpoint prediction?

We first evaluated on different targets the use of multiple instance learning (MIL) on the HANCOCK dataset (Fig. 3A). Among others, we predicted the localization of the tumor (namely: oropharynx, larynx,

and oral cavity) based on the WSI data and superlabels derived from the demographic information (see "Methods"). As only parts of the tissue are normally indicative of the localization, an MIL approach seems appropriate. We rely on the recently introduced Clustering-constrained Attention Multiple Instance Learning (CLAM) concept[20]. We tested different encoding strategies to represent the various input modalities (see "Methods") to determine the effect of the encoding strategy on the endpoint prediction (here: tumor localization). We

found that deep neural networks that were trained self-supervised on histological image data provide embeddings with higher content retaining properties, yielding larger AUC values (0.94 and 0.96 test AUC on average for ResNet18[21] and the UNI encoder) in contrast to supervised models (Fig. 3B). This highlights the use of histopathology foundation models such as UNI[22] also in the context of MIL. To evaluate the pathological validity of this approach, we retrieved the most attended patches for unseen WSIs using the model's attention scores (Fig. 3C, see "Methods"). We found that patches with important morphological details highly related to the origin were deemed important. For example, tissue in the oral cavity should contain a relatively high amount of glands compared to the oropharynx and larynx (Fig. 3C).

Knowing that we can use imaging data to predict relevant targets, we focused on the key strength of HANCOCK - its multimodality. We were interested if the combination of WSI and TMA data could yield improvements in the prediction of patient survival, an important endpoint in clinical practice (Fig. 3D). Our results suggest that indeed, incorporating both WSIs and TMAs improves the patient survival prediction (average test AUC 0.69 on WSI + TMA vs. 0.65 (WSI) and 0.52 (TMA), Fig. 3E). Leveraging the capabilities of CLAM, we can further determine the attention given to the individual data sources using the model's attention scores. We found that especially HE-stained tissues are predictive for patient survival, however, also other modalities and stains have positive attention scores, indicating a complex interaction towards survival prediction (Fig. 3F).

## Discussion

In this study, we provide a monocentric dataset - HANCOCK - comprising 763 patients with multimodal data. The modalities include demographical, pathological, and blood data, WSIs from primary cancer and lymph nodes, and TMAs with IHC staining. We show that the dataset is rich and diverse, and not biased towards a single domain (Figs. 1 and 2). We, in addition, showed in a proof-of-concept study how multimodal MIL can be performed using HANCOCK (Fig. 3), and how important the incorporation of the latest pathology foundation models, such as UNI, is ref. 22. By integrating multimodal data through diverse machine and deep learning approaches, we can show that this allows better prediction of survival and recurrence (Fig. 2F) throughout extracted features (Fig. 2) and fused image sources (Fig. 3). With our transparent and open approach, we hope the HANCOCK dataset will fuel further developments in multimodal data integration and head and neck oncology. By reproducing previous findings, such as the predictive behavior of HPV and PD-L1, we believe that HANCOCK will be very useful in biomarker discovery and validation.

MIL is a wide field for exploration[20,23,24] and has seen recent advances through the incorporation of vision transformers, allowing more generalized histopathology models[22]. With our proof-of-concept study, we laid the foundation for future work (Fig. 3). We have not explored the use of MIL in a full pipeline, including other patient features, which could identify any demographic bias[25] or could be extended to include transcriptomics analysis[26] for improved patch representations.

We further relied consistently on an early fusion approach for training any multimodal AI in our study. This means first fusing the features of distinct modalities and then training a single model, which is recommended as an initial strategy[18]. Future studies should also evaluate if the classification performance could be improved using joint fusion, where neural networks are not only used as feature extractors but are also trained in the process[18]. An in-depth comparison of different methods for extracting and fusing features, especially from our comprehensive histologic image data, could be very beneficial.

We extracted ICD codes from the surgery reports and integrated them into multimodal embeddings using bag-of-words. However, we did not incorporate the plain texts themselves. Since the surgery reports describe the tumor resection in detail and could potentially provide additional information about the severity of the disease, they could be further explored. For example, text embeddings could be extracted using a pre-trained transformer and integrated into the multimodal vectors[27].

Our Machine Learning models are limited to binary classification, however, other options could be explored using the available event data. For example, a regression model could be implemented to predict the time to events such as recurrence or death. Moreover, models could be trained to predict risk scores using a loss function such as Cox partial likelihood loss as proposed by Chen et al.[28].

In this work, the densities of CD3- and CD8-positive cells were computed from the TMAs. We analyzed these regarding their relationship to clinical outcomes (see Supplementary Fig. S8) and integrated them in the multimodal vectors for ML model training (see Fig. 2A). In the future, immune cells expressing the markers CD56, CD163, PD-L1, and MHC-1 as available in HANCOCK could be analyzed as well and integrated for ML model training accordingly.

It has been shown that tissue or cell detection and subsequent classification can enable the investigation of quantitative biomarkers[29,30]. Therefore, annotations of the histologic images in our dataset could be beneficial for biomarker discovery. We already provide manual, board-certified pathologist supervised annotations of tumor regions in the WSIs of the primary tumor. However, these annotations were done sparsely instead of exhaustively. We aim to extend HANCOCK in the future, for example, by creating high-quality annotations of distinct cell types. To this end, we could leverage Deep Learning models and existing manual annotations of nuclei. The annotation or segmentation of larger tissue regions could also be considered and incorporated into the dataset. Further, combining molecular data with histopathological data is a promising approach[31]. Hence, we aim to further integrate genomic or transcriptomic data to increase the long-term impact of the dataset.

Finally, HANCOCK allows the possibility to explore the concept of digital twins, a digital representation of cancer patients, that could improve decisions in cancer care[32]. We implicitly used this concept in the training/test data split (Fig. 2) to compute the cosine similarity between patients to ensure a specific distribution of patients in a given subset (see "Methods").

## Methods
### Data collection
The data was acquired from the Department of Otorhinolaryngology and Head and Neck Surgery and from the Pathological Institute of the University Hospital in Erlangen. All data was collected and published following the local ethics committee vote (#23-22-Br, Ethics Committee of Friedrich-Alexander-University Erlangen-Nürnberg). Retrospective, multimodal data was gathered from patients who were diagnosed with head and neck cancer between 2005 and 2019. The respective ethics committees and institutions waived the need for informed consent as only retrospective data has been used. Only patients who had a curative first treatment were included. The (self-reported) gender was only recorded and had no role in the study design. The modalities in our dataset can be categorized into image data (histopathological images), structured data (clinical, pathological, and blood data), and free text (surgery reports). Supplementary Fig. S11 shows the available and missing data types for all patients.

Tissue samples of the respective patients were collected from the pathological archive of the University Hospital in Erlangen. The samples originate from the primary tumor and, if present, positive lymph nodes that had been resected. The tissue samples had been fixed in formalin, embedded in paraffin, and routinely stained with HE. The 709 primary tumor sections were scanned using a 3DHistech P1000 at 82.44 × magnification and with a resolution of 0.1213 $\frac{\mu m}{pixel}$. A single slide was available for 701 cases, whereas two slides were available for eight

cases. The 396 lymph node sections were scanned using an Aperio Leica Biosystems GT450 at 40 × magnification with 0.2634 $\frac{\mu m}{pixel}$ and using 3DHistech P1000 at 51.42 × magnification with 0.1945 $\frac{\mu m}{pixel}$. All digitized WSIs were stored in the pyramidical Aperio file format (.svs). In addition, TMAs were created from the paraffin-embedded primary tumor blocks. The TMA cores with a diameter of 1.5 mm were extracted from the tumor center and the tumor invasion front. They were stained using HE, and they were stained for specific immune cell populations using the IHC markers CD3, CD8, CD56, CD68, CD163, PD-L1, and MHC-1. CD3-positive cells represent T cells, CD8-positive cells represent cytotoxic T cells, and CD56-positive cells represent natural killer cells. CD68 and CD163 were used to detect monocytes and macrophages. PD-L1 plays a major role in regulating the immune response. It is expressed by tumor cells to deactivate cytotoxic T cells and is a target for immunotherapy[33]. The major histocompatibility complex class I (MHC-1) displays antigens to cytotoxic T cells and is also important for determining the prognosis and treatments involving immunotherapy[34]. From each patient, at least two cores were collected per origin and marker. This resulted in 368 TMAs, each with cores arranged in 12 rows by 6 columns. The TMAs were scanned using a 3DHistech P1000 at 82.44 × magnification with a resolution of 0.1213 $\frac{\mu m}{pixel}$.

Structured pathological data originating from the analysis of the primary tumor and lymph node sections was harmonized and compiled in tabular format. It includes comprehensive information such as the cancer site, staging, grading, and histologic type. The clinical data includes each patient's age, sex, and smoking status. It further contains information and timestamps of events such as treatments, recurrence, progress, metastasis, or death. The data was collected from the hospital information system and by screening various documents such as general and radiotherapy records. Blood test results of the corresponding patients in a range of 14 days around local surgery were retrieved from the hospital's archive. Each measurement was accompanied by the parameter's name, group, unit, and LOINC code (Logical Observation Identifiers Names and Codes)[35].

Surgery reports were collected by filtering the hospital's database by patient identifiers and time range. Reports of patients diagnosed in 2006 were not available, as reports were not entered into the database until 2007. The surgery reports follow a template that includes the medical history and report in the document's body and metadata in the header. All documents were compiled into a .pdf file.

## Data preprocessing
The data was anonymized by assigning a unique, consecutive ID ("001" to "763") randomly to each patient using a custom Python script. Our data is patient-centered. This means that each WSI, each core in a TMA, each surgery report, and each entry in the structured data is mapped to a single patient ID. The preprocessing steps for each data modality are described in the following.

TMAs and WSIs were converted from the manufacturer's file format (.mrxs) to the pyramidical Aperio file format (.svs) using the 3DHistech SlideMaster conversion tool. An Aperio SVS file contains a macro image and a label image. The label image in particular contains potentially identifying information. Therefore, we anonymized the files by removing the label images, i.e., by replacing the image with zeros. To allow the mapping of each TMA core to the corresponding patient, we created TMA maps in CSV (comma-separated values) format (comma-separated values) based on the TMA Grand Master initial mapping that can be imported into QuPath using a custom Python script.

We identified the most important clinical and pathological features and ensured that these were complete for all patients. We performed data cleaning to remove inconsistent or redundant data. For patients with more than one entry in the clinical table, we kept the entry with the earlier diagnosis date. Further entries were removed and merged with the previous entries because they reported a recurrence

of the disease rather than the initial diagnosis (only applicable in five cases). We de-identified the clinical and pathological data by removing all names and dates. The year of the initial diagnosis was retained, but its date was removed. For anonymization purposes, all dates of events were replaced by the number of days since the initial diagnosis. This way, the timeline from the diagnosis to the end of treatment could still be reconstructed. We corrected spelling errors, summarized and harmonized table entries, and assigned self-explanatory labels. The tables were finally converted into JavaScript Object Notation (JSON). Descriptions of all fields in the JSON files with their data types and possible values were summarized in data dictionaries, shown in Supplementary Tables S1, S2, and S3.

The results of blood tests were available as structured, tabular data. We first filtered the data to select values that were measured at specified units, excluding intensive care units. For each patient, we chose a single pre-operative measurement of each parameter. Relevant blood tests were routinely performed one to three days before surgery, or rarely on the morning of the day of surgery. Therefore, the latest available measurement before the surgery date or, if not available, the value from the day of surgery, was selected. The number of available measurements for these time points is shown in Supplementary Fig. S12. The complete blood count, coagulation parameters, electrolytes, and renal function parameters were routinely assessed. Additional parameters were calcium, magnesium, glomerular filtration rate, and glucose. Although it was only available for 94 patients, we included C-reactive protein (CRP) since elevated CRP levels are associated with poor prognosis in patients with head and neck cancer[36]. The blood dataset was converted to JSON format.

The surgery reports were first converted from .pdf to .txt format. Each document had a header containing the operating clinicians, treatment date, the patient's name, and identifiers such as the admission number. The header additionally contained OPS codes and ICD codes. We used regular expressions in Python to search for keywords and obtain relevant data. This way, we extracted ICD codes, OPS codes, and the medical history along with the surgery report itself. We selected reports from the first treatment date, i.e., from the local surgery, and discarded all others. Most patient names had already been masked when they had been entered into the system. However, many texts contained names of operating clinicians. Therefore, we used regular expressions to substitute any names following medical or academic titles. In addition, we performed a search using regular expressions and lists of all names of patients and clinicians. Finally, the reports and medical histories were screened manually for any remaining identifying information. Patient names, clinician names, locations, and dates were replaced by placeholders. The number of replaced terms is shown in Supplementary Table 4. The documents were saved to plain text (.txt) files. In addition, we translated all surgery reports and medical histories from German to English using the DeepL API[37]. For translating short descriptions to English, we used ChatGPT (GPT-3.5)[38]. For convenience, HANCOCK contains the German original and the translated version of the texts. Supplementary Fig. S3 shows word clouds of the most common terms in the translated documents.

## Annotation of primary tumor sections
For training AI models on WSIs using supervised learning, the annotation or segmentation of present tumor regions is usually required[39]. WSIs often contain large areas of tissue that might be irrelevant or even misleading for the corresponding task. We sparsely annotated representative tumor areas in the primary tumor sections using QuPath. To this end, we manually selected one or several regions of interest representing the tumor's histology while avoiding areas that contain artifacts, white background, or healthy tissue such as muscular or glandular tissue. This approach is based on the protocol for the analysis of deep texture representations[40]. An exhaustive annotation of all present tumor regions or distinct tissue types was not possible due to

time constraints. We provide the resulting polygon annotations in ".geojson" format to enable effortless extraction of tumor tiles for future work.

## Multimodal patient vectors

We created multimodal patient vectors for two purposes. First, the vectors were used to determine a dataset split for training and testing. Second, they were used to train models to predict outcomes or treatment choices. To this end, we created embeddings that condensed data from each modality and concatenated them to a single vector per patient.

We encoded the clinical and pathological features using different techniques based on their type. Binary encoding was applied for features such as lymphatic, vascular, or perineural invasion, the patient's sex, or the presence of carcinoma in situ. The pT stage and pN stage were considered ordinal features and transformed into consecutive labels. Categorical features such as primary site or histologic type were assigned labels and were later one-hot encoded. For integrating laboratory parameters, we used the raw values of the hematology group, i.e., the complete blood count.

The ICD codes, extracted from surgery reports, provide a more detailed classification of the disease than the available structured data does. The sequence of ICD codes for each patient was considered a sentence and converted to vectors using a bag-of-words model, inspired by the bag-of-disease-codes approach by Placido et al.[41]. To this end, the first four characters of each ICD code were used. Codes covered by less than three patients were discarded.

The structured pathological data did not contain any information about the immune response of each patient. To include this information, we performed a quantitative analysis of TMAs using the open-source software QuPath (version 0.4.3)[42]. The density of T lymphocytes has been shown to be a prognostic marker[43,44]. Inspired by the Immunoscore[45,46], we computed the density of CD3- and CD8-positive cells in the tumor center and invasion front per tumor area. To this end, we used QuPath to de-array the TMAs and match the tissue cores with patient IDs. Next, tissue detection was performed using thresholding. Strong artifacts were manually removed from the detected regions. Using QuPath's positive cell detection feature, we obtained the positive cell count per mm$^2$ tumor area. Supplementary Fig. S13A shows exemplary TMA cores with detected positive cells, and Supplementary Fig. S13B the respective cell densities. The distribution of the densities is shown in Supplementary Fig. S13C.

The single-modality vectors for each patient were finally concatenated to a multimodal vector with a length of 104. We used UMAP to visualize the multimodal patient vectors in 2D. Beforehand, missing values were replaced by the most frequent value per feature or, in the case of numerical features, by the mean value. Next, one-hot encoding was performed for categorical features and z-score normalization was applied to ordinal and numeric features, i.e., the values were centered around the mean with unit variance. The axes were normalized to the range between zero and one. We used the libraries scikit-learn and umap-learn in Python to preprocess and transform the data.

## Dataset split using a genetic algorithm

We aimed to provide a training dataset and a test dataset that is suitable to test any AI algorithm for its generalizability. We aimed for our test dataset to fulfill the following criteria proposed by Wagner et al.[19]. First, the data should be split at a patient level. Second, both datasets should follow a similar distribution of target classes, in this case, the recurrence and survival status. We created two distinct dataset splits, each into 80% training and 20% test data. The first split should follow the distribution of the training dataset concerning relevant characteristics, by including information from different modalities. The second should be out of distribution and contain outlier cases. To create both splits, we used evolutionary optimization[47].

We implemented a genetic algorithm in Python, where each individual represented a possible split by a vector of zeros (patients assigned to training) and ones (patients assigned to test). The objective of the genetic algorithm was to maximize the fitness of an individual, i.e., of a split with $N$ test points. Before computing the fitness of each split, missing values were imputed, and categorical features were subsequently one-hot encoded. Our imputation strategy was for categorical data, the most frequent value, and for numerical data, the average. A penalty was subtracted from the fitness to achieve a class-balanced split. This penalty was defined as the sum of differences between each class distribution $d = \frac{N_{positive}}{N}$ overall and in the current test dataset. Considering recurrence and survival status as target classes, the number of classes was $C = 2$ in our case. The penalty for $C$ classes was weighted by a weight $\alpha$. A similar approach was introduced by Florez-Revuelta, who used a genetic algorithm to split multi-label data while maximizing the similarity between class distributions[48]. We calculated the fitness of an individual as follows:

For the in-distribution split, the fitness of an individual was defined as the sum of cosine distances from each test point $x_i$ to its nearest neighboring test point $x_{i,nn}$:

$$\text{fitness}_{in} = \sum_{i=1}^{N} \left( 1 - \frac{\vec{x_i} \cdot \vec{x}_{i,nn}}{\left\| \vec{x_i} \right\| \left\| \vec{x}_{i,nn} \right\|} \right) - \alpha \sum_{k=1}^{C} |d_k - d_{k,all}| \quad (1)$$

For the out-of-distribution split, we calculated the sum of cosine distances between all pairs of test points $x$:

$$\text{fitness}_{out} = \sum_{i=1}^{N-1} \sum_{j=i+1}^{N} \left( 1 - \frac{\vec{x_i} \cdot \vec{x_j}}{\left\| \vec{x_i} \right\| \left\| \vec{x_j} \right\|} \right) - \alpha \sum_{k=1}^{C} |d_k - d_{k,all}| \quad (2)$$

The population size was set to 10,000, and the genetic algorithm was terminated after 50 iterations with no further improvement. The population was iteratively updated using parent selection (tournament selection with elitism) and one-point crossover with inversion mutation until convergence. The genetic algorithm was only applied to patients with complete patient vectors. However, for some patients, not all required modalities were available. These were subsequently assigned to the training dataset. The final splits were summarized as a list of patient IDs in JSON format.

## Outcome prediction for distinct dataset splits

For training Machine Learning models to predict recurrence or survival, three different data splits were used. The first split defined "in distribution" cases as test data, the second split defined "out of distribution" data as test data, and the third split defined cases with oropharyngeal cancer as test data (see Fig. 2D). For survival prediction (see Fig. 2E), cases with non-tumor-specific death were excluded. All other cases, including those with unknown causes of death, were considered. The class labels correspond to the survival status, i.e., "living" and "deceased". Binary class labels were also defined for recurrence prediction (see Fig. 2E). The classes were defined as (i) patients who had no recurrence and survived at least three years and (ii) patients who had a recurrence within three years.

As recommended by Huang et al., we applied an early fusion approach as an initial strategy, i.e., we created the multimodal patient vectors and trained a single model[18]. We used three different train-test splits of the dataset, namely the in-distribution and out-of-distribution datasets created using the genetic algorithm. Another split was created by assigning all laryngeal carcinomas to the test dataset. We used the Synthetic Majority Oversampling Technique (SMOTE) to handle class imbalance[49] using the Python package "imbalanced-learn" implementation. We trained Random Forest classifiers using the scikit-learn library in Python and tested it against other available classifiers in the

scikit-learn package, such as Adaboost for boosting algorithms and a support vector classifier (SVC). Similar to previous reports[50,51], we found random forests superior to the other classifiers. The full analysis is shown in Supplementary Tables S6 and S7. One classifier was trained and tested for each of the three splits (see Fig. 2D).

## Multiple instance learning

We used multi-instance learning (MIL) as a weakly supervised method that requires only a slide-level label for a given set of data. Briefly, MIL considers each data unit as a bag of instances and calculates a relevance score for each instance with respect to the bag label[52,53]. In our case, a bag consists of the data modalities (WSI, TMA or combination), and the instances are the tiles from these modalities. The relevance of each instance to the general label is calculated by using the attention-based pooling function to find the attention score on each instance that represents how informative (positive) or uninformative (negative) they are for a given task. Attention scores serve as an interpretability mechanism to highlight which regions (tiles) contribute the most and the least to the slide-level decision. However, because we do not have tile-level importance scores, we do not compute a separate AUC for individual tiles. Instead, the AUC values reported in Fig. 3B, C represent the model's ability to classify entire slides using slide-level ground truth labels as provided by our demographics data (see section "Data Collection").

Clustering-constrained Attention Multiple Instance Learning (CLAM[20]) is a widely used framework that utilizes the MIL paradigm for multiple classes. We followed the recommended approach by using both the bag loss and a clustering loss, which groups instances using pseudo labels (derived from MIL attention scores) to differentiate between positive and negative instances for a given task. We applied CLAM for task localization of the primary tumor within WSIs from samples collected in the oropharynx, larynx, and oral cavity. In an initial assessment, we observed that hypopharyngeal samples exhibited features similar to those of laryngeal and oropharyngeal samples; therefore, we excluded this class from further CLAM analyses. To determine whether there are histologically apparent features that can distinguish the different sampled locations, we analyzed three previously unseen WSIs per class, representing a total of nine patients. Each row within a class corresponds to a different test WSI and consists of a stack of 15 highly attended tiles from that WSI (see also above).

We sampled about 3% of the patches with actual resolution from each WSI with a patch size of 256 × 256 pixels and encoded them through multiple encoders, which were pre-trained in either a supervised or self-supervised manner. We found that this sampling percentage is sufficient for downstream tasks. The encoders with supervised pretraining on ImageNet were ResNet50-A with an embedding size of 1024, ResNet50-B with an embedding size of 2048, EfficientNetB0, and ConvNeXtBase. The encoders with self-supervised pretraining on histology data were ResNet18[21] and UNI, a histology foundation model[22].

We further explored whether MIL can be used to predict patient survival status. To this end, cases with observation periods shorter than 3 years and status "alive" were removed from the dataset, as it is not certain whether their status would remain unchanged with a longer observation period. Furthermore, we force-labeled those whose data observation time had passed 3 years and their status was 'deceased' to 'alive' as the event could not be correlated to the data anymore. In this task, we included multi-stained TMA data for each patient. We achieved this by extracting the cores using QuPath at a downsampling factor of 4 and grouping the cores from differently stained TMAs into an image for each patient. Each TMA core was sampled with about 32 tiles at a 256 × 256 pixel tile size. These images were then fed forward to the CLAM pipeline to segment tissue and retrieve embeddings from the respective encoder. We used all available staining, namely CD163, CD3, CD56, CD68, CD8, HE, MHC-1, and PD-L1. Lastly, we combined WSI and multi-stained TMA data by concatenating the embeddings and training the MIL head. Figure 3E shows the resulting performance. To further analyze which modality contributes most to distinguishing whether a patient will survive, the test data is fed to the CLAM model at inference. The model assigns an attention score to each instance (i.e., each tile), quantifying its contribution to the overall classification decision. We then trace each tile back to its corresponding data modality (e.g., WSI, TMA-CD8, TMA-PD-L1, etc.). This results in the attention score distributions for each modality as shown in Fig. 3F. This visualization reveals which data modalities are most influential in deciding on the classification task.

## Data analysis

Data analysis was performed using established methods available in Python, such as numpy and pandas. Overall survival curves were estimated using the Kaplan-Meier method[54]. The analysis considered the time between the initial diagnosis and death or the end of follow-up. Patients who were alive at the end of the follow-up were censored. We computed overall survival curves for all patients and for patients grouped by different characteristics, see Supplementary Fig. S6.

The clinical data includes various events, such as treatments, progress of the disease, diagnosis of metastases, recurrence, and death or end of follow-up. We visualized the timelines of these events, see Supplementary Fig. S5.

## Statistics and evaluation

The performance of classifiers was reported using ROC curves and corresponding AUC scores computed using Python routines. To compute ROC curves and AUC scores, ML models were either trained and evaluated five times (see Fig. 2E). The ROC curves and AUC scores were then averaged over the iterations or the ten folds, respectively.

We applied the Wilcoxon-Mann-Whitney test to compare the distribution of CD3-positive and CD8-positive cell density of patients grouped by recurrence and survival status, as shown in Supplementary Fig. S8.

## Reporting summary

Further information on research design is available in the Nature Portfolio Reporting Summary linked to this article.

## Data availability

The HANCOCK dataset is publicly available at https://hancock. research.fau.eu/ and will be mirrored to The Cancer Imaging Archive (TCIA) at https://doi.org/10.7937/rcty-5h16. An overview of the dataset, including the number and format of files, is shown in Supplementary Fig. S14. The underlying data plotted in the main Fig. 1–3 is available in the Source Data file. Source data are provided in this paper.

## Code availability

Code is written in Python unless otherwise stated. Code for data exploration, processing histologic images, feature extraction, generating data splits, outcome prediction, and adjuvant treatment prediction is available at https://github.com/ankilab/HANCOCK_MultimodalDataset under an Apache 2.0 license. A frozen version of the code is available at Zenodo[55].

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

## Acknowledgements

This work was funded in part by the Federal Ministry of Education and Research (BMBF) to A.O.G. and M.E. (01KD2211B) and to A.M.K. (01KD2211A).

## Author contributions

M.D.: Data curation, Investigation, Methodology, Software, Visualization, Writing - original draft M.B.: Data curation, Conceptualization, Resources T.H.: Data curation, Resources, Writing - review & editing S.B.: Data curation, Resources, Writing - review & editing H.M.: Investigation, Methodology, Software, Writing - review & editing D.J.F.: Investigation, Methodology, Software, Validation, Writing - review & editing H.K.: Resources, Methodology C.M.: Methodology A.H.: Supervision, Funding acquisition H.I.: Supervision, Funding acquisition M.E.: Conceptualization, Supervision, Resources, Writing - original draft, Project administration, Funding acquisition A.O.G.: Conceptualization, Supervision, Resources, Writing - original draft, Project administration, Funding acquisition A.M.K.: Conceptualization, Supervision, Resources, Writing - original draft, Project administration, Funding acquisition

## Funding

## Competing interests

A.H. declares general disclosures (Honoraria for lectures or consulting/advisory boards for AbbVie, AstraZeneca, Biocartis, BMS, Boehringer Ingelheim, Cepheid, Diaceutics, Gilead, Illumina, Ipsen, Janssen, Lilly, Merck, MSD, Novartis, Pfizer, QUIP GmbH, and other research support from AstraZeneca, Biocartis, Cepheid, Gilead, Illumina, Janssen, Novartis, Owkin, Qiagen, QUIP GmbH). ME declares general disclosures (Personal fees, travel costs, and speaker's honoraria from Zytomed Systems, Merck, Eisai, MSD, AstraZeneca, Janssen-Cilag, Cepheid, Roche, Astellas, Diaceutics, Owkin, BMS, BicycleTX, QuiP GmbH; research funding from AstraZeneca, Janssen-Cilag, STRATIFYER, Cepheid, Roche, Gilead, Owkin, QUIP GmbH, BicycleTX; advisory roles for Ferring, Diaceutics, MSD, AstraZeneca, Janssen-Cilag, GenomicHealth, Owkin, BMS, BicycleTX, Merck; member of the clinical advisory board of BicycleTX; stock ownership: BicycleTX.). All other authors declare no competing interests.
