## [Peer Review File · Nature Communications]

A multimodal dataset for precision oncology in head and neck cancer

Corresponding Author: Dr Andreas Kist

Version 0:

Reviewer comments:

Reviewer #2

(Remarks to the Author)

The authors answered most of the questions, except questions 3 and 8 (which were answered with "N/A" for unclear reasons. These questions don't seem to be related to the paragraphs removed and these sentences still appear)

Regarding the new MIL section added:

- Fig 3B: I don't see clear explanation how the AUC was computed for the localization. Is it per tile, and clarify again what is used as a reference? IT would also be useful to show side by side examples of heatmaps and ground truths.

- Fig 3c: There seems to be 3 "groups" (rows) for each type of tissue. What does it represent, what's the difference between the 3 groups and how were those selected? I wouldn't expect that simple extraction of the "most attended patches" would result naturally in a 3-group split?!

- Fig 3F: I would appreciate more information in the method on how these were obtained. It is unclear what this graph is exactly showing and where it comes from.

As a final side note, I did not receive marked-up version of the manuscript highlighting changes made. Not sure if it was forgotten, but not seeing directly the changes highlighted in the manuscript made the reviewing process more difficult and more time-consuming.

(Remarks on code availability)

Reviewer #3

(Remarks to the Author)

My previous comment to authors was that the manuscript may better fit a journal specializing in methodological issues in medicine, rather than a clinically oriented journal. The authors' response that the manuscript has been submitted to Nature Communications addresses my comment.

(Remarks on code availability)

Version 1:

Reviewer comments:

Reviewer #2

(Remarks to the Author)

The questions were addressed and I don't have any further comments.

(Remarks on code availability)

Rebuttal letter of the revised manuscript Dörrich et al., submitted to Nature Communications

We thank the Editor and the Reviewers for their kind re-evaluation of our manuscript. In the following, we provide a point-by-point response to the specific comments on our manuscript.

Reviewer #1 (Remarks to the Author):

EDITORIAL NOTE: This reviewer was not able to submit their report. However, they were replaced by Reviewer #2, who considers that all concerns were addressed.

We highly appreciate Reviewer #2's feedback on Reviewer #1's comments.

Reviewer #2 (Remarks to the Author):

The authors answered most of the questions, except questions 3 and 8 (which were answered with "N/A" for unclear reasons. These questions don't seem to be related to the paragraphs removed and these sentences still appear)

RESPONSE: In the process of removing and re-working large parts of the manuscript, we may have assigned these questions to the now non-existent adjuvant treatment section. We copied the original questions here (indicated in dark cyan) and respond to your initial comment.

3- Line 343: "Only patients who had a curative first treatment were included." – can you please explicitly state what are the other categories which were removed (and maybe why)?

RESPONSE: Thank you for this important question. In our study, we sought to assemble a population-based, consecutive cohort of patients with HNSCC who presented with localized disease and no evidence of distant metastases. All included patients underwent treatment with curative intent—either surgery alone or surgery in combination with radiotherapy or radiochemotherapy. "Curative intent" implies that the primary tumor and any locoregional lymph node metastases can be completely resected, thereby offering the possibility of a cure. To avoid systematic bias, we excluded patients who already had metastatic disease, as

the assumption for these patients is that the tumor cannot be fully eradicated (i.e., non-curable), necessitating palliative treatment. Because metastatic disease is managed differently and typically has a worse prognosis, we focused exclusively on patients who were initially treated with curative intent (“curative first treatment”).

8- Line 404: “because it reported a recurrence of the disease rather than the initial diagnosis.” – why not keep the recurrence information somewhere? I thought RFS is something some group may be interested in trying to predict?

RESPONSE: We agree with the reviewer. The recurrence information is incorporated and available in the dataset (**clinical_data.json**, entries “*recurrence*” and “*days_to_recurrence*”). We meant that we are removing secondary entries of the same patient, as they can be mixed up with the primary tumor diagnosis leading to biased data. In addition, only five patients were affected by these duplicates. We now clarified this in the main text in lines 352-354. It now reads: “Further entries were removed and merged with the previous entries because they reported a recurrence of the disease rather than the initial diagnosis (only applicable in five cases).”

Regarding the new MIL section added:

- Fig 3B: I don’t see clear explanation how the AUC was computed for the localization. Is it per tile, and clarify again what is used as a reference? IT would also be useful to show side by side examples of heatmaps and ground truths.

RESPONSE: We thank the reviewer for that point and would like to clarify: CLAM is a weakly supervised MIL framework. Therefore, we only have slide-level labels - a tile-level ground truth is thus not available. As a result, the AUC reported in Fig. 3B is computed based on the slide-level classification labels, which is already provided according to the demographics data (same file as in our response to Question #8, namely **clinical_data.json**). In Fig. 3B, these classes correspond to the three sampled locations: larynx, oral cavity, and oropharynx.

The attention scores serve as an interpretability mechanism to highlight which regions (tiles) contribute most to the slide-level decision. However, because we do not have tile-level importance score, we do not compute a separate AUC for individual tiles. Instead, the reported AUC reflects the model’s ability to correctly classify an entire slide based on the aggregated information from multiple tiles.

For multi-class AUC computation, we follow a one-vs-rest approach:

- The AUC is calculated for each class against the rest.
- This is repeated for all class combinations.
- The final reported AUC is the macro-average of these values across all classes.

For further clarity, we added this information in the Methods section on lines 513-523.

We also added the phrase “slide level” to Figure captions 3B and 3E for clarification.

- Fig 3c: There seems to be 3 “groups” (rows) for each type of tissue. What does it represent, what’s the difference between the 3 groups and how were those selected? I wouldn’t expect that simple extraction of the “most attended patches” would result naturally in a 3-group split?!

RESPONSE: We thank the reviewer for this question. We clarified this now in the Methods and the main body of the manuscript. As you correctly pointed out, the simple extraction does not naturally group in any classes.

The weakly-supervised learning task was to classify the super-label “tumor location” derived from the demographic patient information (see answer above). Using multiple instance learning, the AI model should tell us where the tumor was initially located given the bags of tiles: the model’s objective is to classify these images while also identifying which regions within the WSIs contribute most to the classification. To achieve this, the model assigns attention scores to individual tiles within each WSI, quantifying their contribution to the overall classification. Higher attention scores indicate greater relevance in distinguishing between the tumor locations.

In line with typical oropharyngeal tissue and laryngeal tissue, the most attended patches contain tissue-specific, histologically apparent features that a pathologist would use to determine tissue characteristics/tumor location. In Fig. 3c, we visualize the most attended tiles from three test WSIs per class, representing a total of nine patients. Each row within a class corresponds to a different test WSI and consists of a stack of 15 highly attended tiles from that WSI. These selected patches highlight the tiles that the model deemed most informative for classification. This visualization underscores that the model is not making arbitrary decisions but rather focusing on histologically relevant features.

We clarified this in the Methods sections on lines 525-537.

- Fig 3F: I would appreciate more information in the method on how these were obtained. It is unclear what this graph is exactly showing and where it comes from.

RESPONSE: We agree that the attention score was not in-depth explained. We added more information to the Methods (lines 558-567) and commented on this in the Results section (lines 192-216). The Methods text reads: "To further analyze which modality contributes most to distinguishing whether a patient will survive, the test data is fed to the CLAM model at inference. The model assigns an attention score to each instance (i.e., each tile), quantifying its contribution to the overall classification decision. We then trace each tile back to its corresponding data modality (e.g., WSI, TMA-CD8, TMA-PD-L1, etc.). This results in the attention score distributions for each modality as shown in Figure 3F. This visualization reveals which data modalities are most influential in deciding on the classification task."

As a final side note, I did not receive marked-up version of the manuscript highlighting changes made. Not sure if it was forgotten, but not seeing directly the changes highlighted in the manuscript made the reviewing process more difficult and more time-consuming.

RESPONSE: We are very sorry for this inconvenience, this should not have happened. This slipped through in the transfer/submission process on our side. With this revision, we ensured to provide a markup manuscript.

Reviewer #3 (Remarks to the Author):

My previous comment to authors was that the manuscript may better fit a journal specializing in methodological issues in medicine, rather than a clinically oriented journal. The authors' response that the manuscript has been submitted to Nature Communications addresses my comment.

RESPONSE: We thank Reviewer #3 for their final comment on our manuscript.